# Excess prevalence of chronic diseases in elderly people with diabetes and non-diabetics in Poland

Katarzyna Więckiel-Lisowska, Agata Wojciechowska, Łukasz Wierucki, Sonia Woch, Krzysztof Flis, Adrian Lange, Marcin Rutkowski, Tomasz Zdrojewski, Piotr Bandosz*

Department of Preventive Medicine and Education, Medical University of Gdansk, Gdansk, Poland

* piotr.bandosz@gumed.edu.pl

## Abstract

### Objective

Diabetes increases the risk of several chronic conditions. However, their excessive prevalence among older adults with diabetes in Poland is unknown.

### Methods

The prevalence of chronic diseases was assessed in the nationally representative random sample of 5,987 Polish adults aged 60+ (Polsenior2 study, data collected between 2018 and 2020). Each participant's history of hospitalisation due to coronary heart disease (CHD), stroke, and cancer was assessed. Diagnosis of arterial hypertension (AH), cognitive impairment (CI), and chronic kidney disease (CKD) was established based on the questionnaire, blood pressure measurements, Mini-Mental State Examination, and laboratory tests. Diabetes was diagnosed if the participant reported being diagnosed with the disease or their measured HbA1c was ≥ 48 mmol/mol (≥6.5%). Age- and sex-adjusted prevalence ratios of chronic conditions for participants with diabetes versus those without diabetes were calculated using Poisson regression.

### Results

In the multivariate model, the prevalence ratio for CHD history was 1.98 (95%CI: 1.66–2.37), for CKD: 1.90 (95%CI: 1.66–2.18), for stroke: 1.47 (95%CI: 1.15–1.88), for AH: 1.22 (95%CI: 1.17–1.27). Cancer and cognitive impairment prevalence did not differ between people with and without diabetes.

The mean number of chronic diseases was 52% higher in participants with diabetes vs nondiabetic subjects at age 60–69 (1.72 (95%CI: 1.60–1.84) vs. 1.13 (95%CI: 1.07–1.18), respectively). However, this value was only 10% higher in subjects aged 90+ (2.74 (95%CI: 2.45–3.04) vs. 2.49 (95%CI: 2.37–2.62), respectively).

License, which permits unrestricted use, distribution, and reproduction in any medium, provided the original author and source are credited.

**Data availability statement:** The minimal data set is available at https://polsenior2.mug.edu.pl/68966.html.

**Funding:** This paper was implemented under contract No. 6/5/4.2/NPZ/2017/1203/1257 to implement the task in the field of public health of The Operational Objective No. 5 point 4.2. of the National Health Program for the years 2016-2020, entitled "Health Status and Its Socioeconomic Covariates of the Older Population in Poland - the Nationwide PolSenior2 Survey" (PolSenior2).

**Competing interests:** The authors have declared that no competing interests exist.

## Conclusions

Elderly Polish citizens with diabetes suffer more often from coronary heart disease, stroke, chronic kidney disease, and arterial hypertension. The study emphasises that the excess prevalence of chronic diseases among people with diabetes is high in the youngest-old population but diminishes in the oldest-old people.

## Introduction

Type 2 diabetes is considered a non-infectious epidemic disease. Its burden is rising in most of the developed countries. According to WHO data, the global prevalence of diabetes almost doubled between 1980 and 2014, from 4.7% to 8.5% [1]. This increase is driven by population ageing, an increase in obesity and a sedentary lifestyle. The prevalence of diabetes in the adult Polish population is 7.0% and 6.7% in males and females, respectively [2]. It is even more frequent in the elderly, and more than a quarter of the subjects aged 65 suffer from this disease [3].

Diabetes is a powerful and independent risk factor for many chronic conditions, including coronary artery disease (CHD) [4], stroke [5], chronic kidney disease (CKD) [6], arterial hypertension (AH) [7], cognitive impairment (CI) [8], several cancers [9] and others. The excess risk of these diseases in people with diabetes is known from many observational studies. However, to our knowledge, there is no data on the excess prevalence of these conditions in people with diabetes in Poland. This knowledge may be helpful in estimating resources needed for healthcare planning and for modelling studies [10] and also have some clinical usefulness, allowing evaluation of the pre-test probability of these chronic conditions in people with diabetes.

The aim of this paper is to answer the question of how much more prevalent CHD, stroke, CKD, AH, CI and cancer are among elderly subjects with diabetes in Poland in comparison to the part of the population without diabetes.

## Materials and methods

To answer the study question, we used the data collected during the Polsenior2 project, a nationwide survey of the older Poles' overall health status, quality of life, and socioeconomic situation.

The PolSenior2 study was approved by the Independent Bioethics Committee for Scientific Research at Gdańsk University of Medical Sciences (approval number: NKBBN/257/2017 dated July 3, 2017). The approval covered the recruitment of participants, fieldwork (surveys, tests, measurements), as well as the collection and storage of biological material. Each participant signed a written informed consent form for participation in the study. Additionally, each participant was assigned a unique identification number to maintain the confidentiality of personal data collected during the study.

### Study sample

The study sample consisted of 5,987 males and females aged 60+, randomly drawn from the general Polish population, using multistage stratified and clustered sampling.

The detailed design of the study is described elsewhere [11]. In short, a three-stage sampling procedure was used. During the first stage, Poland was divided into 78 territorial strata based on region (voivodeship) and municipality size. The number of respondents to draw in each stratum was defined proportionally to the size of its population aged 60 + . Then, 137 municipalities out of 3,118 were randomly selected, also with probability proportional to their elderly population size. In the second stage of the sampling procedure, streets and villages were drawn in urban and rural territories, respectively, with the probability proportional to their population size at age 60 + . Finally, during the third stage, individual respondents were randomly selected within 5-year age strata, aged 60–64, 65–69, 70–74, 75–79, 80–84, 85–89, 90 and older. The sampling frame for individual respondents was the national registry of Polish citizens (PESEL database, which is a complete registry of country citizens). For each age group, a similar number of males and females were intended to draw, resulting in an overrepresentation of the oldest part of the population. This increased statistical power for separate analyses in the very elderly. Overrepresentation was corrected during the poststratification process using weights to obtain estimates for all Polish citizens aged 60 + . For each respondent, another nine subjects of similar age, sex and place of living were drawn. These "spare" addresses were used only in case of unavailability of the primary respondent. The only inclusion criteria were: age 60 and above and the availability of patients at the address registered in the PESEL registry. The only exclusion criterium was the lack of respondents' consent to participate in the study. Examination of patients at their homes and minimizing the exclusion criteria were intended to avoid, as much as possible, any selection bias toward the healthier part of the populations.

Study visits were performed by trained nurses at respondents' homes. During these visits, questionnaires were used to collect information about health and social issues. Several measurements were taken, i.e. blood pressure, weight and height. Cognitive function was assessed using Mini-Mental State Examination (MMSE). Finally, fasting venous blood samples were taken, serum was separated and frozen in local laboratories and finally tested in a central laboratory. The study protocol was approved by the local bioethics committee.

## Diagnosis of diabetes

Diabetes was considered present in all participants who reported having been diagnosed with the disease, regardless of their actual measured blood glucose or HbA1c levels. Among participants unaware of their diagnosis, diabetes was identified if the measured HbA1c was ≥ 48 mmol/mol (≥6.5%). This definition is consistent with current clinical guidelines [4,12]. There were missing data on HbA1c measurements in 18% of study subjects. In those cases, diabetes was diagnosed if fasting serum glucose concentration was ≥ 126 mg/dL. The reason for the relatively high number of missing data on HbA1c was that this sample was the last one from multiple taken. In the case of trouble with collecting enough blood, the specimen for HbA1c measurement was abandoned. HbA1c was measured in blood using the HPLC method (Biorad), and fasting glucose level was measured in plasma using enzymatic methods (hexokinase, Siemens).

## Diagnosis of chronic diseases

Arterial hypertension status was evaluated according to current European clinical guidelines [13]. AH was diagnosed if measured systolic blood pressure during any of two separate visits was ≥ 140 mmHg or diastolic blood pressure was ≥ 90 mmHg. The participant was also assumed to have hypertension if he/she declared taking any antihypertensive drugs during two weeks preceding the interview and declared being diagnosed with hypertension. Blood pressure measurements were taken at the participant's home using validated, automatic oscillometric devices. History of CHD hospitalisation, stroke and cancer was assessed based on participant's or their informant's declaration. CKD was diagnosed if the eGFR measured during the study was lower than 60 mL/min or the urine albumin/creatinine ratio was ≥ 30 mg/g. Cognitive impairment was diagnosed if the subject's MMSE score, corrected for age and education [14], was < 27. We used the version of questionnaire which was validated previously for Polish population [15].

The MMSE may be difficult to administer or yield unreliable results due to various conditions unrelated to cognitive function, such as impaired hearing, speech, vision, or motor function. In our study, we did not administer the MMSE to subjects with severe hearing impairment. If verbal communication with the patient was impossible and the family reported end-stage dementia, we assigned an MMSE score of zero. For participants with other disabilities, such as vision or motor impairments, the MMSE score was adjusted to reflect the fraction of the maximum possible score based on the patient's capabilities.

### Statistical analysis

We compared the fractions of study subjects with a history of CHD, stroke, or cancer and the prevalence of diagnosed AH, CKD and CI between participants with or without diabetes. The significance of these differences was evaluated using the chi-square test, with a significance threshold set at $p < 0.05$. The results were presented stratified by sex and age groups. The statistical analysis accounted for the complex survey design. Overrepresentation of the older age group was corrected during poststratification using weights based on current official population data.

As age and sex were the main confounders in the analysis, we performed a multivariate analysis using Poisson regression. The primary model ("Model 1") included only demographic variables: sex and age as covariates. Sensitivity analyses were conducted by sequentially adding additional variables to the model. These included socioeconomic status (education and municipality size) in "Model 2," cardiovascular risk factors (smoking and total cholesterol) in "Model 3," and BMI in "Model 4."

Education was assessed via questionnaire and categorized into four levels: primary or incomplete primary, middle school, secondary or post-secondary, and higher education. Municipality size was determined based on the official national administrative database (TERYT) and categorized as rural, urban with <50,000 inhabitants, urban with 50,000–200,000 inhabitants, and urban with >200,000 inhabitants. BMI was calculated using height and weight measurements taken during the study visit by a nurse, using standardized equipment. Smoking status was self-reported in the study questionnaire. Fasting total cholesterol levels were measured using the enzymatic method in a central laboratory.

Finally, we calculated the average number of chronic diseases in part of the population with and without diabetes (including a history of hospitalisation due to CHD, stroke, cancer, arterial hypertension, chronic kidney disease and cognitive impairment). Results were presented by age using LOESS regression.

Analysis was conducted in R (R Core Team, 2021) [16] using *survey* package [17] and the figure was produced using the package *ggplot2* [18].

### Results

The final sample size was 5,987 subjects out of 10,635 eligible, resulting in a response rate of 56.3%. Sample characteristics are presented in Table 1. Due to the intentional overrepresentation of the older segment of the population in the sample, both the unweighted average age and unweighted prevalence of the studied conditions are higher than the weighted data. This is because the risk of these diseases is strongly associated with age.

The overall prevalence of diabetes was 25.3% (95%CI: 22.7–27.9) in males and 21.5% (95%CI: 19.4–23.7) in females. The highest morbidity was observed in females in their eighth decade (32.0%) and males in their seventh decade of life (28.7%).

For all country citizens aged 60 +, a history of CHD hospitalisation was reported in 13.6% (95%CI: 12.3–14.8), history of stroke in 8.0% (95%CI: 7.0–9.0), any cancer in 8.9% (95%CI: 7.8–10.0). The prevalence of arterial hypertension was 74.4% (95%CI: 72.3–76.4), CKD 17.7% (95%CI: 16.5–19.0) and cognitive impairment 19.0% (95%CI: 16.1–22.0).

Some of the considered chronic diseases were more prevalent in participants with diabetes than those without diabetes. We found a higher proportion of CHD hospitalisation history (23.5% vs 10.8%, p<0.001), stroke (11.2% vs 6.9%, p<0.001), CKD (30.9% vs 13.8%, p<0.001), AH (88.1% vs 70.5%, p<0.001) and CI (19.0% vs 16.0%, p=0.05). We found no excess prevalence of cancer (9.0% vs 8.9%, ns) among subjects with diabetes (Table 2).

**Table 1. Characteristics of the study sample.**

|  | Unweighted sample | Weighted sample* |
|---|---|---|
| **Demographic variables** | **Percent or mean (95%CI)** | **Percent or mean (95%CI)** |
| Percentage of females | 51.1 (49.8–52.3) | 58.2 (58.2–58.2) |
| Age:60–69 | 34.2 (33–35.4) | 54.6 (54.6–54.6) |
| 70–79 | 32.8 (31.6–34) | 27.9 (27.9–27.9) |
| 80–89 | 24.4 (23.3–25.5) | 14.9 (14.9–14.9) |
| 90+ | 8.6 (7.9–9.3) | 2.6 (2.6–2.6) |
| Mean age | 75 (74.7–75.2) | 70.7 (70.6–70.8) |
| **Prevalence of chronic conditions** |  |  |
| Diabetes | 25.1 (24.0–26.2) | 23.1 (21.3–25.0) |
| History of CHD hospitalisation | 17.0 (16.0–17.9) | 13.6 (12.3–14.8) |
| Stroke | 9.1 (8.4–9.8) | 8.0 (7.0–9.0) |
| Cancer | 10.3 (9.5–11.1) | 8.9 (7.8–10.0) |
| Arterial hypertension | 77.2 (76.2–78.3) | 74.7 (73.1–76.2) |
| Cognitive impairment** | 37.9 (36.6–39.1) | 32.8 (30.4–35.1) |
| Chronic kidney disease | 26.0 (24.8–27.1) | 17.7 (16.5–19.0) |

*weights were calculated to correct age and sex distribution to the population of Polish elderly citizens in 2020

**MMSE<27 (using Mungas correction)

In multivariate analysis, we found about two times higher age- and sex-adjusted prevalence of CHD history (1.98 (95%CI: 1.66–2.37) in participants with diabetes. For existing CKD, this ratio was 1.90 (95%CI: 1.66–2.18); for a history of stroke, 1.47 (95%CI: 1.15–1.88); for arterial hypertension, 1.22 (95%CI: 1.17–1.27). We found no significant age- and sex-adjusted excess prevalence of cancer history (PR: 0.98 (95%CI: 0.78–1.23)) nor CI (1.05 (95%CI: 0.93–1.19)). Additional adjustments for socioeconomic covariates resulted in virtually unchanged results. However, further adjustments for other risk factors (smoking, total cholesterol, body mass index (BMI)) resulted in weaker observed differences between both groups for most studied exposures. With these adjustments, the higher prevalence of a history of stroke was no longer observed, and the excess prevalence of a history of CHD was two times smaller. Still, the effect on chronic kidney disease prevalence remained unchanged (Table 3).

*Model 1 – adjusted for demographic variables only: age (and additionally for sex in the "All" group)*

*Model 2 – like model 1 + socioeconomic status: size of municipality and education (and additionally for sex in "All" group)*

*Model 3 – like model 2 + cardiovascular risk factors: smoking and total cholesterol (and additionally for sex in the "All" group)*

*Model 4 – like model 3 + BMI (model 4 was included separately, as the BMI has known strong relation to the risk of diabtes).*

The average number of chronic diseases in people with diabetes aged 60 was about 50% higher than in their counterparts without diabetes. The number of diseases was increasing with age in both groups. Still, this increase was less steep in people with diabetes, resulting in an insignificant difference between people with and without diabetes in the ninth decade of life (Table 4 and Fig 1).

## Discussion

We found that the probability of having a history of CHD or having chronic kidney disease is almost twice as high in elderly people with diabetes as in their counterparts without. This excess prevalence is less pronounced for stroke and arterial hypertension, where the probability of having these conditions is 47% and 22% higher, respectively. On the other hand,

**Table 2. The estimated prevalence of chronic conditions among people with diabetes and without diabetes in the studied population.**

| Condition | Sex/Age | People with diabetes | People without diabetes | p-value |
|---|---|---|---|---|
| History of hospitalisation due to CHD | All | 10.8 (9.4–12.1) | 23.5 (20.0–27.0) | <0.001 |
| | Females | 8.2 (6.7–9.6) | 20.3 (15.8–24.7) | <0.001 |
| | Males | 14.5 (12.3–16.8) | 27.3 (22.5–32.0) | <0.001 |
| | 60–69 | 7.8 (5.9–9.8) | 20.2 (14.9–25.4) | <0.001 |
| | 70–79 | 12.5 (10.3–14.8) | 26.6 (21.4–31.8) | <0.001 |
| | 80–89 | 20.2 (16.7–23.8) | 26.3 (20.3–32.3) | 0.08 |
| | 90+ | 12.5 (7.6–17.5) | 22.1 (10.6–33.5) | 0.11 |
| History of stroke | All | 6.9 (5.8–7.9) | 11.2 (8.8–13.6) | <0.001 |
| | Females | 5.9 (4.7–7.1) | 10.3 (7.3–13.3) | <0.001 |
| | Males | 8.3 (6.5–10.2) | 12.3 (8.7–15.8) | 0.036 |
| | 60–69 | 4.5 (3.4–5.7) | 8.8 (5.3–12.4) | 0.01 |
| | 70–79 | 7.6 (5.5–9.7) | 13.9 (9.6–18.2) | 0.003 |
| | 80–89 | 14.3 (11.2–17.4) | 11.9 (7.9–16.0) | 0.368 |
| | 90+ | 14.4 (7.4–21.4) | 13.5 (3.8–23.3) | 0.883 |
| History of cancer | All | 8.9 (7.7–10.1) | 9.0 (7.0–11.0) | 0.923 |
| | Females | 9.3 (7.5–11.0) | 9.7 (7.0–12.3) | 0.782 |
| | Males | 8.4 (7.0–9.8) | 8.2 (5.3–11.2) | 0.924 |
| | 60–69 | 6.9 (5.6–8.3) | 8.9 (5.5–12.2) | 0.225 |
| | 70–79 | 12.2 (9.9–14.4) | 8.5 (5.9–11.1) | 0.026 |
| | 80–89 | 10.7 (8.2–13.3) | 9.5 (6.1–13.0) | 0.607 |
| | 90+ | 10.7 (5.9–15.4) | 14.6 (6.3–22.9) | 0.36 |
| Arterial hypertension | All | 70.5 (68.6–72.3) | 88.1 (85.8–90.3) | <0.001 |
| | Females | 69.9 (67.7–72.1) | 88.3 (84.9–91.7) | <0.001 |
| | Males | 71.2 (68.1–74.4) | 87.8 (84.2–91.5) | <0.001 |
| | 60–69 | 64.2 (61.4–67.0) | 88.0 (84.2–91.7) | <0.001 |
| | 70–79 | 77.6 (74.6–80.5) | 90.2 (86.8–93.7) | <0.001 |
| | 80–89 | 81.6 (78.2–85.0) | 85.0 (79.6–90.4) | 0.245 |
| | 90+ | 81.6 (76.4–86.7) | 85.9 (78.2–93.5) | 0.281 |
| Chronic kidney disease | All | 13.8 (12.6–15.0) | 30.9 (27.1–34.7) | <0.001 |
| | Females | 13.7 (12.0–15.4) | 33.4 (28.7–38.1) | <0.001 |
| | Males | 13.9 (11.7–16.1) | 27.9 (22.1–33.7) | <0.001 |
| | 60–69 | 5.5 (3.8–7.1) | 16.8 (11.1–22.6) | <0.001 |
| | 70–79 | 14.9 (12.5–17.3) | 32.9 (27.9–37.9) | <0.001 |
| | 80–89 | 39.0 (34.5–43.5) | 55.0 (48.0–62.0) | <0.001 |
| | 90+ | 65.9 (59.3–72.5) | 75.3 (63.4–87.2) | 0.2 |
| Cognitive impairment | All | 31.1 (28.7–33.6) | 35.4 (31.0–39.7) | 0.058 |
| | Females | 31.8 (28.9–34.7) | 35.6 (30.4–40.8) | 0.155 |
| | Males | 30.2 (27.2–33.2) | 35.1 (29.8–40.3) | 0.085 |
| | 60–69 | 24.4 (21.5–27.3) | 30.6 (24.5–36.7) | 0.031 |
| | 70–79 | 33.1 (29.2–37.1) | 35.1 (29.4–40.8) | 0.576 |
| | 80–89 | 49.3 (44.9–53.6) | 42.5 (35.0–50.0) | 0.094 |
| | 90+ | 70.8 (64.0–77.6) | 69.5 (57.7–81.2) | 0.818 |

Table 3. Prevalence ratio of chronic conditions in people with diabetes compared to people without diabetes. Poisson regression (PR) analysis. Separate PR models are presented, with adjustments to. Model 1: demographic variables only. Model 2: additionally, socioeconomic status. Model 3: additionally cardiovascular risk factors except BMI, Model 4 additionally BMI. Model 4 was presented separately, as the BMI is known to be strongly realeated to the risk of diabetes.

| Condition | Prevalence ratio | | | |
|---|---|---|---|---|
| | Model 1 | Model 2 | Model 3 | Model 4 |
| **All** | | | | |
| History of CHD | 1.98 (1.66–2.37) | 1.95 (1.64–2.33) | 1.54 (1.28–1.85) | 1.52 (1.25–1.86) |
| History of stroke | 1.47 (1.15–1.88) | 1.41 (1.10–1.81) | 1.16 (0.88–1.52) | 1.10 (0.83–1.47) |
| History of cancer | 0.98 (0.78–1.23) | 1.00 (0.79–1.25) | 0.98 (0.77–1.23) | 0.98 (0.77–1.25) |
| AH | 1.22 (1.17–1.27) | 1.22 (1.17–1.27) | 1.17 (1.13–1.22) | 1.10 (1.06–1.14) |
| Cognitive impairment | 1.05 (0.93–1.19) | 1.02 (0.90–1.15) | 1.01 (0.89–1.15) | 1.05 (0.92–1.21) |
| CKD | 1.90 (1.66–2.18) | 1.87 (1.63–2.15) | 1.82 (1.58–2.08) | 1.76 (1.53–2.03) |
| **Males** | | | | |
| History of CHD | 1.82 (1.44–2.29) | 1.78 (1.41–2.25) | 1.26 (0.98–1.61) | 1.20 (0.92–1.55) |
| History of stroke | 1.42 (1.00–2.02) | 1.39 (0.97–1.98) | 1.14 (0.77–1.68) | 1.13 (0.76–1.66) |
| History of cancer | 0.95 (0.66–1.39) | 0.94 (0.65–1.37) | 0.98 (0.67–1.43) | 1.00 (0.68–1.47) |
| AH | 1.23 (1.15–1.31) | 1.24 (1.16–1.32) | 1.18 (1.11–1.26) | 1.13 (1.06–1.20) |
| Cognitive impairment | 1.13 (0.96–1.33) | 1.12 (0.95–1.32) | 1.12 (0.94–1.33) | 1.15 (0.95–1.39) |
| CKD | 1.90 (1.47–2.46) | 1.91 (1.48–2.47) | 1.84 (1.39–2.43) | 1.83 (1.37–2.44) |
| **Females** | | | | |
| History of CHD | 2.21 (1.72–2.83) | 2.20 (1.72–2.81) | 1.92 (1.51–2.44) | 2.01 (1.57–2.58) |
| History of stroke | 1.52 (1.11–2.08) | 1.49 (1.10–2.03) | 1.22 (0.90–1.66) | 1.12 (0.80–1.56) |
| History of cancer | 0.99 (0.74–1.34) | 1.03 (0.77–1.37) | 0.98 (0.71–1.34) | 0.96 (0.68–1.36) |
| AH | 1.22 (1.16–1.28) | 1.21 (1.15–1.27) | 1.16 (1.11–1.21) | 1.08 (1.03–1.13) |
| Cognitive impairment | 0.99 (0.85–1.16) | 0.95 (0.82–1.11) | 0.95 (0.81–1.11) | 0.99 (0.83–1.17) |
| CKD | 1.89 (1.63–2.21) | 1.83 (1.57–2.14) | 1.78 (1.53–2.07) | 1.72 (1.45–2.03) |

Table 4. The average number of chronic conditions (including: a history of hospitalisation due to CHD, stroke, cancer, arterial hypertension, chronic kidney disease, cognitive impairment).

| Age group: | The average number of chronic conditions (95% CI) | | | |
|---|---|---|---|---|
| | Age 60–69 | Age 70–79 | Age 80–89 | Age 90+ |
| People without diabetes | 1.13 (1.07–1.18) | 1.57 (1.51–1.64) | 2.13 (2.02–2.24) | 2.49 (2.37–2.62) |
| People with diabetes | 1.72 (1.60–1.84) | 2.08 (1.99–2.17) | 2.27 (2.11–2.42) | 2.74 (2.45–3.04) |
| % higher in people with diabetes vs those without diabetes | 52.2% | 32.5% | 6.6% | 10.0% |

we found no excess prevalence of CI and history of cancer. The higher prevalence of a history of CHD, stroke and hypertension in people with diabetes is partly diminished after adjustments for total cholesterol, smoking and BMI.

It may seem surprising that we found no excess prevalence of CI or history of cancer among people with diabetes, as there is plenty of existing evidence linking diabetes to increased risk of these conditions [9,19]. The potential explanation can be higher case fatality related to diabetes among people with cancer or dementia. Prevalence of the disease is the result of the complex relationship between incidence, case fatality, remission rate and mortality [20]. Diabetes may influence more than only incidence. For example, suppose case fatality among cancer participants with diabetes is higher than in those without diabetes. In that case, we can expect a lower number of subjects with a cancer history in the population, as they are dying at a higher rate. This can annihilate (at least partially) the effect of increased risk of diabetes

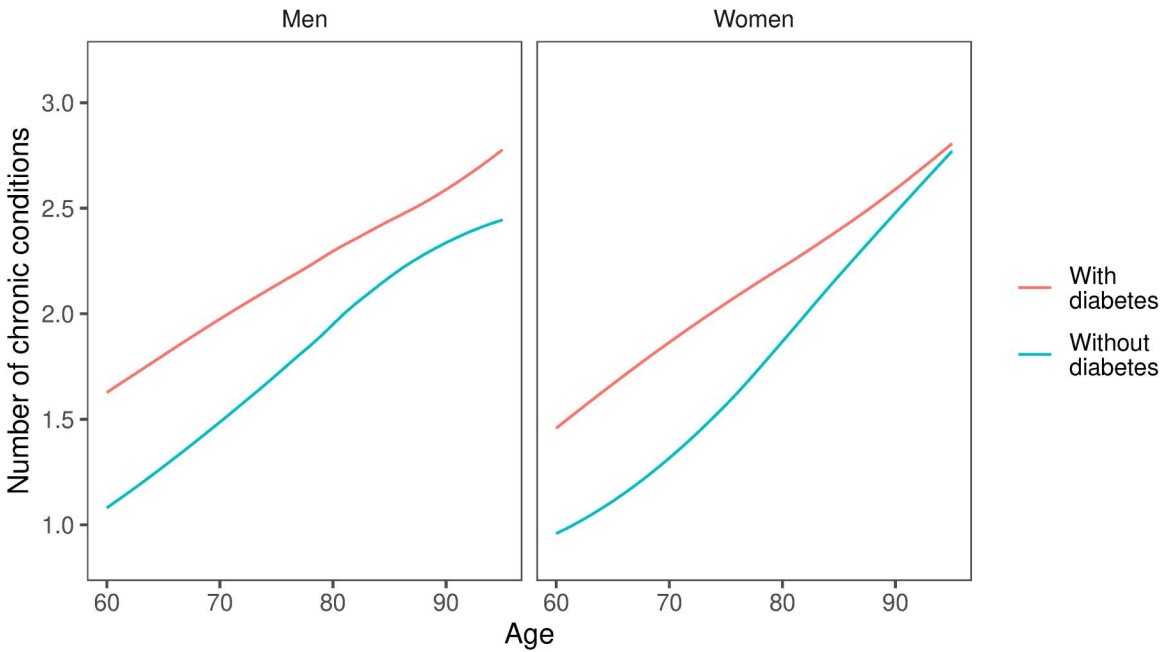

**Fig 1. The average number of chronic conditions among Polish males and females with diabetes and without diabetes aged 60 +.** Included conditions are a history of hospitalization due to CHD, stroke, cancer, arterial hypertension, chronic kidney disease, and cognitive impairment. LOESS regression. Shaded areas represent local 95% confidence limits.

on observed prevalence. In fact, there is evidence supporting a much worse prognosis after incident cancer if diabetes coexists [21,22]. Similarly, for dementia, participants suffering from this condition since middle age, with a ≥ 15-year history of diabetes, were dying almost twice as fast as those without diabetes, according to an Australian study [23].

The differences in survival may also explain lower prevalence ratios for CHD and stroke than usually reported hazard ratios or relative risks coming from longitudinal studies. Diabetes is a well-established risk factor for CHD and stroke. In the middle-aged population, the risk of incident stroke is reported to be about 1.4–3 times higher if diabetes exists [24]. These numbers are even higher in the case of estimates unadjusted to other cardiovascular risk factors, like in the study by Tuomilehto, where males with diabetes were at a sixfold increased risk of death from stroke, and the risk for females with diabetes was 8.2 times higher [25]. The risk of CHD is usually reported to be increased twofold in males and threefold in females [26]. It is important to note that these relative risks or hazard ratios are becoming lower with the increasing age of the studied population.

Another interesting finding is the diminishing difference in the average number of chronic conditions between populations with diabetes and without diabetes in older ages (Fig 1). This can also be potentially explained by poorer survival to older ages of subjects with diabetes accompanied by multimorbidity being (at least partly) caused by macro- and micro-vascular complications of diabetes.

## Study strengths and limitations

The major strength of our study is a nationally representative and relatively large sample. This allowed us to provide estimates for the general Polish population with reasonable precision. The problem of selection bias towards healthier elderly subjects was minimised by performing all examinations at participants' homes and allowing informants to provide information if the study participant was unable to do it due to disability. Another strength is using diabetes, arterial hypertension

and chronic kidney disease definitions based on current clinical guidelines, covering both diagnosed and undiagnosed cases.

Our analysis also has several limitations. First, the response rate of 56.3% may be worse than in some previous studies. This is the common problem of the recent population surveys, where due to social changes response rate exceeding 50% is rarely achieved, particularly in the eldest part of the population [27]. Another limitation is the lack of HbA1c measurements in 18% of the sample. This problem is partially overridden by using fasting glucose to assess diabetes status in these subjects. In our study, we use a history of CHD hospitalisation rather than the objective diagnosis of CHD, and a history of cancer and stroke was self-reported only. This study does not take into account the duration of diabetes within participants.

## Conclusions

Elderly Polish citizens with diabetes have CKD and a history of CHD twice as often and a history of stroke and hypertension 47% and 22% (respectively) more often compared with their counterparts without diabetes. However, we found no excess prevalence of cognitive impairment or a history of cancer.

Our study emphasises that the excess prevalence of chronic diseases among people with diabetes is high in the youngest-old population but diminishes in the oldest-old population, where we observed no significant differences between both studied subgroups.

This does not necessarily mean that diabetes is not a risk factor in the very elderly, as it can possibly result from survival bias. Our findings implicate instead that from a public health perspective, the burden of chronic diseases among the oldest-old part of the population is not significantly higher in people with diabetes. This is contrary to the younger part of the population, where the excess prevalence of chronic diseases is markedly higher.

### A bulleted novelty statement

- We assessed the excess prevalence of several chronic diseases in subjects with diabetes aged 60+ in Poland versus the non-diabetic part of the population.

- We found that people with diabetes aged 60 and older suffer more often from coronary heart disease, stroke, chronic kidney disease, and arterial hypertension than those without diabetes. However, we didn't find a higher prevalence of cognitive impairment and a history of cancer.

- The excessive morbidity of studied chronic diseases among people with diabetes is high in the youngest-old population but diminishes in the oldest-old.

## Author contributions

**Conceptualization:** Katarzyna Więckiel-Lisowska, Piotr Bandosz.

**Data curation:** Katarzyna Więckiel-Lisowska, Piotr Bandosz.

**Formal analysis:** Katarzyna Więckiel-Lisowska, Piotr Bandosz.

**Funding acquisition:** Piotr Bandosz.

**Investigation:** Katarzyna Więckiel-Lisowska, Tomasz Zdrojewski.

**Methodology:** Piotr Bandosz.

**Project administration:** Katarzyna Więckiel-Lisowska, Tomasz Zdrojewski, Piotr Bandosz.

**Resources:** Katarzyna Więckiel-Lisowska, Piotr Bandosz.

**Software:** Piotr Bandosz.

**Supervision:** Katarzyna Więckiel-Lisowska, Tomasz Zdrojewski, Piotr Bandosz.

**Validation:** Katarzyna Więckiel-Lisowska, Agata Wojciechowska, Łukasz Wierucki, Tomasz Zdrojewski, Piotr Bandosz.

**Visualization:** Katarzyna Więckiel-Lisowska, Piotr Bandosz.

**Writing – original draft:** Katarzyna Więckiel-Lisowska, Tomasz Zdrojewski, Piotr Bandosz.

**Writing – review & editing:** Katarzyna Więckiel-Lisowska, Agata Wojciechowska, Łukasz Wierucki, Sonia Woch, Krzysztof Flis, Adrian Lange, Marcin Rutkowski, Tomasz Zdrojewski, Piotr Bandosz.

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
