## [Decision Letter · Decision Letter 0]

8 Oct 2024

PONE-D-24-24690Excess prevalence of chronic diseases in elderly people with diabetes in Poland.PLOS ONE

Dear Dr. Więckiel-Lisowska,

Thank you for submitting your manuscript to PLOS ONE. After careful consideration, we feel that it has merit but does not fully meet PLOS ONE’s publication criteria as it currently stands. Therefore, we invite you to submit a revised version of the manuscript that addresses the points raised during the review process.

 Please elaborate more details about the measurement tools  particularly about cognitive  test and also defining  threshold  criteria

We look forward to receiving your revised manuscript.

Kind regards,

Hamid Reza Baradaran, M.D., Ph.D.,

Academic Editor

PLOS ONE

Journal requirements: When submitting your revision, we need you to address these additional requirements. 1. Please ensure that your manuscript meets PLOS ONE's style requirements, including those for file naming. The PLOS ONE style templates can be found at https://journals.plos.org/plosone/s/file?id=wjVg/PLOSOne_formatting_sample_main_body.pdf and https://journals.plos.org/plosone/s/file?id=ba62/PLOSOne_formatting_sample_title_authors_affiliations.pdf 2. We note that the grant information you provided in the ‘Funding Information’ and ‘Financial Disclosure’ sections do not match.  When you resubmit, please ensure that you provide the correct grant numbers for the awards you received for your study in the ‘Funding Information’ section. 3. Thank you for stating the following financial disclosure:  [This paper was implemented under contract No. 6/5/4.2/NPZ/2017/1203/1257 to implement the task in the field of public health of The Operational Objective No. 5 point 4.2. of the National Health Program for the years 2016-2020, entitled "Health Status and Its Socioeconomic Covariates of the Older Population in Poland - the Nationwide PolSenior2 Survey" (PolSenior2)].  Please state what role the funders took in the study.  If the funders had no role, please state: ""The funders had no role in study design, data collection and analysis, decision to publish, or preparation of the manuscript."" If this statement is not correct you must amend it as needed. Please include this amended Role of Funder statement in your cover letter; we will change the online submission form on your behalf. 4. Thank you for stating the following in the Acknowledgments Section of your manuscript: [This paper was implemented under contract No. 6/5/4.2/NPZ/2017/1203/1257 to implement the task in the field of public health of The Operational Objective No. 5 point 4.2. of the National Health Program for the years 2016-2020, entitled "Health Status and Its Socioeconomic Covariates of the Older Population in Poland - the Nationwide PolSenior2 Survey" (PolSenior2).]We note that you have provided funding information that is not currently declared in your Funding Statement. However, funding information should not appear in the Acknowledgments section or other areas of your manuscript. We will only publish funding information present in the Funding Statement section of the online submission form. Please remove any funding-related text from the manuscript and let us know how you would like to update your Funding Statement. Currently, your Funding Statement reads as follows:  [This paper was implemented under contract No. 6/5/4.2/NPZ/2017/1203/1257 to implement the task in the field of public health of The Operational Objective No. 5 point 4.2. of the National Health Program for the years 2016-2020, entitled "Health Status and Its Socioeconomic Covariates of the Older Population in Poland - the Nationwide PolSenior2 Survey" (PolSenior2)].  Please include your amended statements within your cover letter; we will change the online submission form on your behalf. 5. Please include a caption for figure 1. 6. We note that you have indicated that there are restrictions to data sharing for this study. For studies involving human research participant data or other sensitive data, we encourage authors to share de-identified or anonymized data. However, when data cannot be publicly shared for ethical reasons, we allow authors to make their data sets available upon request. For information on unacceptable data access restrictions, please see http://journals.plos.org/plosone/s/data-availability#loc-unacceptable-data-access-restrictions.  Before we proceed with your manuscript, please address the following prompts: a) If there are ethical or legal restrictions on sharing a de-identified data set, please explain them in detail (e.g., data contain potentially identifying or sensitive patient information, data are owned by a third-party organization, etc.) and who has imposed them (e.g., a Research Ethics Committee or Institutional Review Board, etc.). Please also provide contact information for a data access committee, ethics committee, or other institutional body to which data requests may be sent. b) If there are no restrictions, please upload the minimal anonymized data set necessary to replicate your study findings to a stable, public repository and provide us with the relevant URLs, DOIs, or accession numbers. Please see http://www.bmj.com/content/340/bmj.c181.long for guidelines on how to de-identify and prepare clinical data for publication. For a list of recommended repositories, please see https://journals.plos.org/plosone/s/recommended-repositories. You also have the option of uploading the data as Supporting Information files, but we would recommend depositing data directly to a data repository if possible. Please update your Data Availability statement in the submission form accordingly.

Additional Editor Comments:

Please elaborate more details about the measurement tools particularly about cognitive test and also defining threshold criteria

Reviewers' comments:

Reviewer's Responses to Questions

**Comments to the Author**

1. Is the manuscript technically sound, and do the data support the conclusions?

Reviewer #1: Partly

2. Has the statistical analysis been performed appropriately and rigorously? 

Reviewer #1: Yes

3. Have the authors made all data underlying the findings in their manuscript fully available?

Reviewer #1: Yes

4. Is the manuscript presented in an intelligible fashion and written in standard English?

Reviewer #1: Yes

5. Review Comments to the Author

Reviewer #1: this is an important study in terms of policy making in Poland. I think a number of revisions may be necessary:

Title: OK, however, for a better understating of the readers it is better to show the comparator (non-diabetics) in the title as well.

Abstract: Please show the confidence intervals of the estimated prevalences shown for the 2 age groups (60-69 vs 90+) in lines 34-35.

Keywords: OK

Introduction: Ok

Material and method: Please notice to the following issues in this section:

- It is necessary to define the how ethical issues have been dealt with in more detail.

- Please define the define the inclusion and exclusion criteria of the recruited participants in detail and explicitly in this section.

- Please define how patients with diagnosed diabetes and controlled blood sugar were classified. This question arise since the diagnosis of diabetes is based on HbA1C in the paper.

- I think it is necessary to define the time of diagnosis for the proposed chronic disease. Were they diagnosed before the diagnosis of diabetes or after that?

- It is necessary to express how reliability of measurements including cognitive status were measured. This has to be measured and stated in your own study.

- Please express the statistical significancy threshold considered in the study.

- Please define the criteria of including variable for the different models expressed in table 3.

- It is necessary to define the variables used for modelling shown in table 3 as well.

Results: I think the method of results already shown in table 3 and 4 are not addressed in the previous section. On the other hand, I think the format conveyed “The average number of chronic conditions” in each group is not an appropriate one.

Discussion: I noticed for a number of variables shown in table 1, the difference between weighted and unweighted estimates was high. In that case, I think it is necessary to address this finding in this section.

References: OK; some revisions may be necessary considering the accepted format of the journal.

6. PLOS authors have the option to publish the peer review history of their article (what does this mean? ). If published, this will include your full peer review and any attached files.

**Do you want your identity to be public for this peer review?** For information about this choice, including consent withdrawal, please see our Privacy Policy .

Reviewer #1: **Yes: ** Babak Eshrati

---

## [Author Response · Author response to Decision Letter 1]

5 Jan 2025

Dear Reviewers,

The list below is a necessary response to the revision of the manuscript submitted to PLOS ONE titled "Excess Prevalence of Chronic Diseases in Elderly People with Diabetes in Poland."

Firstly, we would like to address the journal's requirements. We have ensured that our manuscript adheres to PLOS ONE's style requirements. Additionally, we have corrected issues related to funding in the "Funding Information" and "Financial Disclosure" sections.

We declare that:

"This paper was implemented under contract No. 6/5/4.2/NPZ/2017/1203/1257 to implement the task in the field of public health of The Operational Objective No. 5 point 4.2. of the National Health Program for the years 2016-2020, entitled 'Health Status and Its Socioeconomic Covariates of the Older Population in Poland - the Nationwide PolSenior2 Survey' (PolSenior2). The funders had no role in study design, data collection and analysis, decision to publish, or preparation of the manuscript."

We kindly request that you update the online submission form on our behalf. Additionally, we have removed the funding information from the "Acknowledgments" section.

We have included a caption for Figure 1, which reads as follows:

"The average number of chronic conditions among Polish males and females with diabetes and without diabetes aged 60+. Included conditions are a history of hospitalization due to CHD, stroke, cancer, arterial hypertension, chronic kidney disease, and cognitive impairment. LOESS regression. Shaded areas represent local 95% confidence limits."

Another issue that required revision was the data availability. According to your suggestion, the minimal data set is available at https://polsenior2.mug.edu.pl/68966.html.

In accordance with the editor's and reviewers' suggestion, we have provided more detailed information in the manuscript about the measurement tools, particularly concerning the cognitive tests, and we have also defined the threshold criteria.

In response to the reviewer’s comments, we have made changes in accordance with their suggestions.

Reviewer #1: this is an important study in terms of policy making in Poland. I think a number of revisions may be necessary:

Title: OK, however, for a better understating of the readers it is better to show the comparator (non-diabetics) in the title as well.

We have changed the title to include non-diabetics. (2 line)

Abstract: Please show the confidence intervals of the estimated prevalences shown for the 2 age groups (60-69 vs 90+) in lines 34-35.

Thank you, we added that detail at this section: „The mean number of chronic diseases was 52% higher in participants with diabetes vs nondiabetic subjects at age 60–69 (1.72 (95%CI: 1.60–1.84) vs. 1.13 (95%CI: 1.07–1.18), respectively). However, this value was only 10% higher in subjects aged 90+ (2.74 (95%CI: 2.45–3.04) vs. 2.49 (95%CI: 2.37–2.62), respectively).” (lines 42-43)

Keywords: OK Thank you.

Introduction: Ok Thank you.

Material and method: Please notice to the following issues in this section:

- It is necessary to define the how ethical issues have been dealt with in more detail.

Thank you, we updated that detail. Now it sounds: “The PolSenior2 study was approved by the Independent Bioethics Committee for Scientific Research at Gdańsk University of Medical Sciences (approval number: NKBBN/257/2017 dated July 3, 2017). The approval covered the recruitment of participants, fieldwork (surveys, tests, measurements), as well as the collection and storage of biological material. Each participant signed a written informed consent form for participation in the study. Additionally, each participant was assigned a unique identification number to maintain the confidentiality of personal data collected during the study.” (lines 77-83).

- Please define the deąfine the inclusion and exclusion criteria of the recruited participants in detail and explicitly in this section.

Thank you, we updated relevant part of the Material and method section. Now it sounds: “The only inclusion criteria were: age 60 and above and the availability of patients at the address registered in the PESEL registry. The only exclusion criterium was the lack of respondents’ consent to participate in the study. Examination of patients at their homes and minimizing the exclusion criteria were intended to avoid, as much as possible, any selection bias toward the healthier part of the population.” (lines 103-107)

- Please define how patients with diagnosed diabetes and controlled blood sugar were classified. This question arise since the diagnosis of diabetes is based on HbA1C in the paper.

Thank you for spotting this lack of precsion. We updated relevant section which now is: “Diabetes was considered present in all participants who reported having been diagnosed with the disease, regardless of their actual measured blood glucose or HbA1c levels. Among participants unaware of their diagnosis, diabetes was identified if the measured HbA1c was ≥48 mmol/mol (≥6.5%).” (lines 115-118)

- I think it is necessary to define the time of diagnosis for the proposed chronic disease. Were they diagnosed before the diagnosis of diabetes or after that?

Unfortunately, the information about the time of diagnosis was not recorded in the questionnaire for most of the chronic diseases analyzed in our study. Even if such data were available, determining the exact onset of diabetes would still remain challenging. This is particularly true for type 2 diabetes, which constitutes the majority of cases and is often diagnosed after a substantial delay due to the absence of symptoms in its early stages. Consequently, in a cross-sectional study like ours, it is virtually impossible for most individuals to ascertain which condition occurred first: diabetes or other chronic diseases.

We agree that establishing the temporal sequence of exposure and effect is crucial when aiming to identify causal relationships. However, seeking causality was not the objective of our study. This causal link has already been demonstrated in numerous other studies. Our cross-sectional design is far less suitable for this purpose compared to longitudinal studies.

Our goal was to determine whether older adults with diabetes exhibit a similar excess prevalence of chronic diseases as observed in middle-aged adults. Having the country representative sample allowed us to generalize results to whole country population. Interestingly, we found that this overrepresentation significantly diminishes among the very elderly. In this age group, the likelihood of having other chronic diseases does not differ much between individuals with and without diabetes. This phenomenon was somewhat counterintuitive to some of us.

- It is necessary to express how reliability of measurements including cognitive status were measured. This has to be measured and stated in your own study.

In our study, cognitive impairment status was assessed using the Mini-Mental State Examination (MMSE) test, which is widely employed as a screening tool for cognitive impairment in both clinical and epidemiological settings. We used a version of the test that had been previously validated for the Polish population and is commonly utilized in clinical practice. Unfortunately, no additional reliability studies were conducted during our study. In the revised version of manuscript, we provide a reference for the validation study in the manuscript.

We think there is also some lack of precise information about using MMSE in the study. We updated methods section with paragraph cited below:

“We used the version of questionnaire which was validated previously for Polish population.

The MMSE may be difficult to administer or yield unreliable results due to various conditions unrelated to cognitive function, such as impaired hearing, speech, vision, or motor function. In our study, we did not administer the MMSE to subjects with severe hearing impairment. If verbal communication with the patient was impossible and the family reported end-stage dementia, we assigned an MMSE score of zero. For participants with other disabilities, such as vision or motor impairments, the MMSE score was adjusted to reflect the fraction of the maximum possible score based on the patient's capabilities.” (lines 137-145)

- Please express the statistical significancy threshold considered in the study.

Thank you for spotting this issue. We updated “Statistical analysis” section with information that we have used significance threshold set at p<0.05. (lines 148-150)

- Please define the criteria of including variable for the different models expressed in table 3.

Thank you for noting the lack of information.

Our first model, “Model 1,” included only demographic variables, as age and sex are the most important confounders in our study.

The second model, “Model 2,” incorporated additional potential socioeconomic confounders available in our data, such as place of residence (represented by municipality) and level of education.

The final two models, “Model 3” and “Model 4,” also included known cardiovascular disease (CVD) risk factors. BMI was added as a separate variable in Model 4, as it is an important risk factor for diabetes. However, including BMI made very little difference to the results.

We updated “Material and method” section and the Figure 3 caption with the information presented above. (lines 154-167)

- It is necessary to define the variables used for modelling shown in table 3 as well.

Thank you, this is updated in the revised version of the manuscript, lines 212-227.

Results: I think the method of results already shown in table 3 and 4 are not addressed in the previous section. On the other hand, I think the format conveyed “The average number of chronic conditions” in each group is not an appropriate one.

Now the “Material and method” section is updated to cover these results.

We agree that the “average number of chronic conditions” may not be ideal and is also quite an abstract measure. We included this as a measure of multimorbidity to present the main finding of the study — the diminishing excess multimorbidity in elderly individuals with diabetes as age increases — in a simple (and graphical) way. An alternative approach would involve a much less readable table with distributions of chronic conditions stratified by age and diabetes status. After internal discussion, we would prefer to keep this approach if possible. (lines 176-179)

Discussion: I noticed for a number of variables shown in table 1, the difference between weighted and unweighted estimates was high. In that case, I think it is necessary to address this finding in this section.

We addressed/explained this in the beginning of the “Results” section. The reason for huge differences was intentional overrepresentation of the eldest subjects in the study, aimed to allow enough statistical power for comparison in older age groups. As the studied diseases are strongly related to age, we observe huge differences between unweighted and weighted sample (as the weights are correcting age structure of the sample to entire country population).

References: OK; some revisions may be necessary considering the accepted format of the journal.

Thank you, we have made the revisions according to the accepted format of the journal.

Kind regards,

Katarzyna Więckiel-Lisowska

---

## [Decision Letter · Decision Letter 1]

30 Jan 2025

Excess prevalence of chronic diseases in elderly people with diabetes and non-diabetics in Poland.

PONE-D-24-24690R1

Dear Dr. Więckiel-Lisowska,

We’re pleased to inform you that your manuscript has been judged scientifically suitable for publication and will be formally accepted for publication once it meets all outstanding technical requirements.

Kind regards,

Hamid Reza Baradaran, M.D., Ph.D.,

Academic Editor

PLOS ONE

Additional Editor Comments (optional):

Reviewers' comments:

Reviewer's Responses to Questions

**Comments to the Author**

1. If the authors have adequately addressed your comments raised in a previous round of review and you feel that this manuscript is now acceptable for publication, you may indicate that here to bypass the “Comments to the Author” section, enter your conflict of interest statement in the “Confidential to Editor” section, and submit your "Accept" recommendation.

Reviewer #1: All comments have been addressed

2. Is the manuscript technically sound, and do the data support the conclusions?

Reviewer #1: Yes

3. Has the statistical analysis been performed appropriately and rigorously? 

Reviewer #1: Yes

4. Have the authors made all data underlying the findings in their manuscript fully available?

Reviewer #1: Yes

5. Is the manuscript presented in an intelligible fashion and written in standard English?

Reviewer #1: Yes

6. Review Comments to the Author

Reviewer #1: thanks to distinguished authors and I appreciate their response, almost all of the comments are addressed. I think the paper is ready to be published.

7. PLOS authors have the option to publish the peer review history of their article (what does this mean? ). If published, this will include your full peer review and any attached files.

**Do you want your identity to be public for this peer review?** For information about this choice, including consent withdrawal, please see our Privacy Policy .

Reviewer #1: **Yes: ** Babak Eshrati

---

## [Editor Report · Acceptance letter]

PONE-D-24-24690R1

PLOS ONE

Dear Dr. Więckiel-Lisowska,

I'm pleased to inform you that your manuscript has been deemed suitable for publication in PLOS ONE. Congratulations! Your manuscript is now being handed over to our production team.

Kind regards,

on behalf of

Professor Hamid Reza Baradaran

Academic Editor

PLOS ONE